# ResiDual: Transformer with Dual Residual Connections

## Abstract

Transformer networks have become the preferred architecture for many tasks due to their state-of-the-art performance. However, the optimal way to implement residual connections in Transformer, which are essential for effective training, is still debated. Two widely used variants are the Post-Layer-Normalization (Post-LN) and Pre-Layer-Normalization (Pre-LN) Transformers, which apply layer normalization after each residual block's output or before each residual block's input, respectively. While both variants enjoy their advantages, they also suffer from severe limitations: Post-LN causes gradient vanishing issue that hinders training deep Transformers, and Pre-LN causes representation collapse issue that limits model capacity. In this paper, we propose ResiDual, a novel Transformer architecture with Pre-Post-LN (PPLN), which fuses the connections in Post-LN and Pre-LN together, and inherits their advantages while avoids their limitations. We conduct both theoretical analyses and empirical experiments to verify the effectiveness of ResiDual. Theoretically, we prove that ResiDual has a lower bound on the gradient to avoid the vanishing issue due to the residual connection from Pre-LN. Moreover, ResiDual also has diverse model representations to avoid the collapse issue due to the residual connection from Post-LN. Empirically, ResiDual outperforms both Post-LN and Pre-LN on several machine translation benchmarks across different network depths and data sizes.

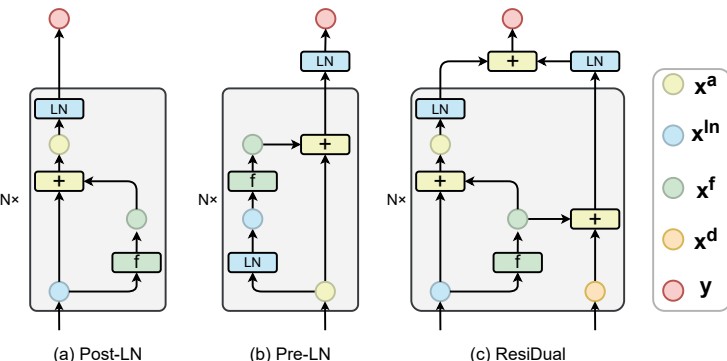

Figure 1: Overview of Post-LN, Pre-LN, and ResiDual. Circles with different colors represent different variables and rectangles represent different operations. See Section 2 for more details.

## 1 Introduction

Transformer (Vaswani et al., 2017) has emerged as a powerful neural network architecture that has been successfully applied in various AI tasks, including machine translation (Vaswani et al., 2017), language modeling and generation (Radford et al., 2018; 2019; Brown et al., 2020), image recognition (Dosovitskiy et al., 2020), and speech synthesis (Ren et al., 2019). Despite its success, researchers are still exploring ways to further enhance its performance and deepen the understanding of its inner workings (Wang et al., 2019; Katharopoulos et al., 2020; Fedus et al., 2021). Among

them, one area of ongoing research is the study of residual connections in the Transformer architecture (Liu et al., 2020; Xiong et al., 2020; Bachlechner et al., 2021). Two variants of residual connections have been proposed since the introduction of the Transformer, known as Post-LN and Pre-LN. The Post-LN variant applies layer normalization (LN) operations after the output of each residual block. This variant is used in several prominent models such as BERT (Devlin et al., 2018), RoBERTa (Liu et al., 2019), and ALBERT (Lan et al., 2019). The Pre-LN variant, on the other hand, applies LN operations before the input to each residual block. This variant is used in models such as the GPT series, ViT (Dosovitskiy et al., 2020), and PaLM (Chowdhery et al., 2022).

Although both variants have been widely used, each one has its own drawbacks, which are summarized in Table 1. As shown in Figure 1, the key difference between the two residual variants is how the layer normalization (LN) normalized the outputs of each block. With Post-LN, the output of lower blocks (i.e., the blocks close to input) are normalized multiple times. As a result, the gradient norm decays exponentially with depth and eventually vanishes in the lower layers (Xiong et al., 2020). This problem does not exist in Pre-LN because the gradient can flow directly to each block. However, the Pre-LN architecture has the representation collapse issue (Liu et al., 2020), which will negatively impact the model's capacity. The representation

| Method | Gradient Vanishing | Representation Collapse |
|---|---|---|
| Post-LN | 🙁 | 😄 |
| Pre-LN | 😄 | 🙁 |
| ResiDual | 😄 | 😄 |

Table 1: Comparison of Post-LN, Pre-LN, and our method. 😄 means the model does not suffers from the issue and 🙁 means the model has such issue.

collapse issue refers to the fact that the hidden representation of higher blocks (i.e., the blocks close to output) will be similar to each other in Pre-LN models. Therefore, the higher blocks will have little contribution to the model capacity.

Several approaches have been proposed to address these problems, which can generally be categorized into three categories. Firstly, some methods aim to modify the architecture, such as DLCL (Wang et al., 2019), NormFormer (Shleifer et al., 2021), RealFormer He et al. (2021), and B2T (Takase et al., 2022), which adds extra components such as aggregations or LNs to stable training. Secondly, some methods add different weights to the residual, such as Admin (Liu et al., 2020), DeepNet (Wang et al., 2022a), $\tau$-ResNet (Zhang et al., 2022), and ReZero (Bachlechner et al., 2021). Lastly, some methods use better initialization, such as T-Fixup (Huang et al., 2020), DeepNet (Wang et al., 2022a), and Foundation Transfomer (Wang et al., 2022b), to reduce variance and stabilize training.

In this study, we focus on the first category and propose a new architecture for Transformer models to address the drawbacks of both variants while retaining their benefits. Figure 1(c) provides an overview of our method. Our design goal is to maintain the advantages of both variants and avoid their disadvantages by employing two residual connections. In particular, our ResiDual model utilizes a Pre-Post-LN (PPLN) that consists two residuals: one is similar to the Pre-LN to prevent the gradient vanishing issue, while the other one akin to the Post-LN, which sustains representation diversity to avoid the representation collapse issue.

To validate the effectiveness of our proposed method, we conduct both theoretical analysis (Section 3) and empirical study (Section 4) to show that our method can achieve the best of both worlds. From the theoretical perspective, we first show that the gradient vanishing is still a critical problem even using Adam (Kingma & Ba, 2014) optimizer. We also show that ResiDual has a bounded gradient-norm thus do not have such an issue. Furthermore, we study the representation collapse issue and show that ResiDual has the same hidden representation diversity as Post-LN. Therefore, ResiDual do not have the representation collapse issue in Pre-LN.

Empirically, we conduct comprehensive experiments on machine translation tasks, which are among the most representative tasks in natural language processing. Our dataset comprises small-scale (IWLST), mid-scale (WMT), and large-scale (OPUS) datasets. Our experimental results demonstrate that our method outperforms baselines across all three datasets.

In summary, this work makes the following contributions:

- We present ResiDual, a simple yet potent variation of the Transformer architecture, which tackles both the gradient vanishing problem in Post-LN and the representation collapse issue in Pre-LN Transformer models.

- Our theoretical analysis demonstrates that this new design can leverage the strengths of both variants while avoiding their weaknesses.

- Our experimental results provide further evidence of the effectiveness of our approach, as it achieves superior performance compared to both the Post-LN and Pre-LN Transformer models across multiple datasets.

## 2 METHOD

### 2.1 DISADVANTAGES OF POST-LN AND PRE-LN

In this section, we briefly review the architecture of Post-LN and Pre-LN, whose illustrations are available in Figure 1 (a) and (b). We will also discuss the shortcomings of each architecture.

**Gradient Vanishes with Post-LN.** The Post-LN architecture is shown in Figure 1 (a). To be more specific, given a Post-LN Transformer network with $N$ residual blocks, we assume the input shape is $n \times d$ where the $n, d$ denotes the sequence length and embedding size[1]. The variables with vector arrow (e.g., $\overrightarrow{\boldsymbol{x}} \in \mathbb{R}^{n \times d}$) denote the whole sequence and the variables without it (e.g., $\boldsymbol{x} \in \mathbb{R}^d$) denote an element of the sequence. We use $\overrightarrow{\boldsymbol{x}}^a \in \mathbb{R}^{n \times d}$ denote the tensor after **a**dd operation and use subscript $k$ (i.e. $\overrightarrow{\boldsymbol{x}}_k^a$) denote the tensor in the $k$-th block. We also use $\overrightarrow{\boldsymbol{x}}_k^{ln} \in \mathbb{R}^{n \times d}$ denotes the normalized tensor and $\overrightarrow{\boldsymbol{x}}_k^f \in \mathbb{R}^{n \times d}$ denotes the output of the function $f_k(\cdot; \boldsymbol{w}_k)$ in the $k$-th block. The $f_k$ can be a self-attention, cross-attention, or feed-forward with parameter $\boldsymbol{w}_k$. Using these notations, the Post-LN computation of each element in the $k$-th block is

$$\boldsymbol{x}_k^a = \boldsymbol{x}_k^{ln} + \boldsymbol{x}_k^f = \boldsymbol{x}_k^{ln} + f_k(\overrightarrow{\boldsymbol{x}}_k^{ln}; \boldsymbol{w}_k); \quad \boldsymbol{x}_{k+1}^{ln} = \mathrm{LN}(\boldsymbol{x}_k^a).$$

Finally, the output $\boldsymbol{y}$ is computed by $\boldsymbol{y} = \boldsymbol{x}_{N+1}^{ln} = \mathrm{LN}(\boldsymbol{x}_N^a)$. Intuitively, the $\boldsymbol{x}_k^f$ is normalized $N - k$ times, so does the gradients of $\boldsymbol{w}_k$. Therefore, the gradients of lower blocks will be small. From Xiong et al. (2020), we know that for Post-LN Transformer, the gradient norm decreases exponentially from deep layers to shallow layers. Intuitively, such an imbalanced gradients will impede the model training. Therefore, in practise, training tricks such as learning-rate warm-up are necessary to train a Post-LN model.

**Representation Collapses with Pre-LN.** With the same notations, the Pre-LN computation is

$$\boldsymbol{x}_k^{ln} = \mathrm{LN}(\boldsymbol{x}_k^a); \quad \boldsymbol{x}_{k+1}^a = \boldsymbol{x}_k^a + \boldsymbol{x}_k^f = \boldsymbol{x}_k^a + f_k(\overrightarrow{\boldsymbol{x}}_k^{ln}; \boldsymbol{w}_k).$$

Similarly, the model output is $\boldsymbol{y} = \mathrm{LN}(\boldsymbol{x}_{N+1}^a) = \mathrm{LN}(\sum_{k=1}^N \boldsymbol{x}_k^f)$. Intuitively, as the $\boldsymbol{x}_k^f$ is only normalized once when computing the $\boldsymbol{y}$, neither the forward nor the backward pass are blocked by LN. Thus, Pre-LN do not have the gradient vanish issue. However, it has another issue called representation collapse. More specifically, Liu et al. (2020) show that the $\frac{\sqrt{\mathrm{Var}[\boldsymbol{x}_k^f]}}{\sqrt{\mathrm{Var}[\boldsymbol{x}_k^a + \boldsymbol{x}_k^f]}}$ is likely to be smaller for higher blocks (i.e, blocks with larger $k$). This means the output of the later blocks ($\boldsymbol{x}_k^f$) has little contribution to the total variance of $\boldsymbol{x}_k^a$. In Section 3.2, we show that the difference between $\boldsymbol{x}_{k+1}^{ln}$ and $\boldsymbol{x}_k^{ln}$ (i.e., $|\boldsymbol{x}_{k+1}^{ln} - \boldsymbol{x}_k^{ln}|$) decays along with $k$, which indicates the input of the higher blocks will collapse to similar values. We also show that this issue may limit the capacity of the model.

### 2.2 RESIDUAL

The goal of our model is to take the advantages of both variants and avoid the both disadvantages. To achieve this goal, we use residuals from both variants and the overview of our method is in

---

[1]We omit the batch dimension that will not affect our analysis.

Figure 1 (c). More specifically, the two residual connections are illustrated in the left and right vertical lines in the Figure. The left one, which is similar to the conventional Post-LN, is

$$\boldsymbol{x}_k^a = \boldsymbol{x}_k^{ln} + \boldsymbol{x}_k^f = \boldsymbol{x}_k^{ln} + f_k(\overrightarrow{\boldsymbol{x}}_k^{ln}; \boldsymbol{w}_k); \quad \boldsymbol{x}_{k+1}^{ln} = \text{LN}(\boldsymbol{x}_k^a).$$

Meanwhile, the right residual, which is similar to the conventional Pre-LN, is formulated by

$$\boldsymbol{x}_{k+1}^d = \boldsymbol{x}_k^d + \boldsymbol{x}_k^f,$$

where $\boldsymbol{x}^d \in \mathbb{R}^{n \times d}$ is the tensor to denote **d**ual residual that similar to $\boldsymbol{x}^a$ in the Pre-LN that allows the gradients directly flow to each block.

Finally, the output $y$ is computed by adding the representation of both residuals, which is

$$\boldsymbol{y} = \boldsymbol{x}_{N+1}^{ln} + \text{LN}\left(\boldsymbol{x}_{N+1}^d\right).$$

### 2.3 DISCUSSION

In this section, we will only introduce the intuitive understanding of ResiDual and the mathematical analysis is provided in Section 3.

**Avoiding the Gradient Vanishing** In ResiDual, gradient of each block flows from both residual connections. Thus, even if the gradient comes from the Post-LN-like residual vanishes, there will still be gradients from the Pre-LN-like residual. This prevents the gradient vanishing issue. We provide the details of the lower-bound of the gradient norm in Section 3.1.

**Avoiding the Representation Collapse** Our Pre-LN-like residual only affects the model output and does not affect the input to each block. Therefore, the representation capacity is the same as a Post-LN model. Furthermore, because the final output of our model is the sum of two residual connections, the representation of the output will not collapse either. We provide the details of the lower-bound of the representation capacity in Section 3.2.

## 3 THEORETICAL ANALYSIS OF RESIDUAL

In this section, we formally study the gradient vanishing and representation collapse issue. We also prove that our method does not have such issues.

### 3.1 THE GRADIENT VANISHING ISSUE

In order to present the analysis in a concise way, we study a simple setting and make several assumptions. In Transformer, the $f$ function can be either a feed-forward block or a multi-head attention block. For a feed-forward block, $f(\boldsymbol{x}) := \boldsymbol{W}\boldsymbol{x}$ where we ignore the layer index. For a multi-head attention block, we have weight matrices $\boldsymbol{W}_Q, \boldsymbol{W}_K, \boldsymbol{W}_V$. For simplicity, we focus on single-head attention. Similar to Xiong et al. (2020), we initialize $\boldsymbol{W}_Q$ to be zero matrices and consequently, the attention is a uniform distribution at initialization and $f(\boldsymbol{x}^{(i)}) := \frac{1}{n}\sum_{j=1}^{n} \boldsymbol{x}^{(j)}\boldsymbol{W}_V$ where we drop the layer index and $\boldsymbol{x}^{(j)}, j \in [n]$ are the input sequence with length $n$. We usually drop the superscript index $^{(j)}$ for notation simplicity when the context is clear itself. We introduce $\overrightarrow{\boldsymbol{x}} := \{\boldsymbol{x}^{(j)}, j \in [n]\}$ and use $\boldsymbol{w}$ to denote the collection of parameter matrices in $f$.

Based on above assumption, without loss of generality, we further assume that the $f$ function keeps the norm, i.e., $\|f(\boldsymbol{x})\| = \|\boldsymbol{x}\|$. This assumption is asymptotically true when the network width goes to infinity and the initialization variance is properly scaled. We assume that the signal is standardized after layer normalization, i.e., $\|\boldsymbol{x}_k^{ln}\| = \sqrt{d}$ for all $k \in [N]$, and that for $\boldsymbol{x} \in \mathbb{R}^d$, the Jacobian matrix through LN satisfies $\frac{\partial \text{LN}(\boldsymbol{x})}{\partial \boldsymbol{x}} \approx \frac{\sqrt{d}}{\|\boldsymbol{x}\|_2}\boldsymbol{I}$. This approximation can be achieved if the mean of $\boldsymbol{x}$ is 0 and the variance is $\frac{1}{d}\|\boldsymbol{x}\|^2$ while ignoring the gradient back-propagated through mean and variance. The rationale in this assumption is that the error signal (gradients) back-propagating through LN becomes smaller as the norm of the input to the LN gets larger. In the Post-LN Transformer, the scale of the inputs to the layer normalization is independent of $N$, and thus the gradients of parameters in the last layer are independent of $N$.

**Gradient Norm Estimation for Post and Pre-LN Transformer.**  From Xiong et al. (2020), we know that for Post-LN Transformer, the gradient norm of the block $k$ decreases exponentially as block index $k$ gets smaller. This indicates that the gradient of the block close to input would be exponentially small for deep transformers. In contrast, for Pre-LN Transformer, the gradient norm of each block is roughly independent with the block index $k$.

For completeness, we rephrase the result from Xiong et al. (2020) with our notations and assumptions. We also present the proof in a more accurate way in Appendix.

**Theorem 3.1** (Gradients of the $k$-th block in the Post-LN and Pre-LN Transformers). *Given the above assumptions on $f$ and LN, for the Post-LN Transformer with $N$ blocks, the gradient of the parameters of the $k$-th block satisfies*

$$\left\|\frac{\partial \mathcal{L}}{\partial \boldsymbol{w}_k}\right\|_F \approx \mathcal{O}\left((1/2)^{(N-k)/2} e^{\sqrt{N-k}}\right), \tag{1}$$

*for the Pre-LN Transformer with $N$ blocks, the gradient of the parameters of the $k$-th block satisfies*

$$\left\|\frac{\partial \mathcal{L}}{\partial \boldsymbol{w}_k}\right\|_F \approx \mathcal{O}\left(\sqrt{\frac{\log(N-k)}{N}}\right), \tag{2}$$

*where we ignore the terms irrelevant with $k, N$.*

**Analysis of Adam**  In practice, adaptive optimizers such as Adam are widely used to train Transformer networks. However, the vanished gradients issue cannot be solved by adaptive optimizers and thus we aim to fix the issue in the network architecture. More specifically, we show that the Adam updates is *ill-conditioned* in vanished gradients. More specifically, let the $\alpha, t, \epsilon, \beta_1, \beta_2$ denote the learning rate, step, smoothing factor, first decay rate and second decay rate, respectively, and the $\boldsymbol{w}^{(t)}, \mathbf{g}, \hat{\mathbf{m}}^{(t)}, \hat{\mathbf{v}}^{(t)}$ denote the parameters, gradients, bias-corrected first and second moment estimation at time t. Meanwhile, we use $\mathbf{u}(\mathbf{g}^{(t)}) = \alpha \cdot \hat{\mathbf{m}}^{(t)}/(\sqrt{\hat{\mathbf{v}}^{(t)}} + \epsilon)$ denote the Adam update (i.e., $\mathbf{w}^{(t)} \leftarrow \mathbf{w}^{(t-1)} - \mathbf{u}(\mathbf{g}^{(t)})$) and the full formula is in Appendix B. Because the Adam update is element-wise, we also use $u(g)$ to denote the scalar function of $\mathbf{u}(\mathbf{g})$, which means $\mathbf{u}(\mathbf{g}) = [u(g_1), u(g_2), \cdots, u(g_d)]$. Then, we will show that, when the gradients vanish, the $\mathbf{u}(\mathbf{g})$ is sensitive to small perturbation (i.e., ill-conditioned) because of its large condition number.

**Theorem 3.2.** *The Adam update function $\mathbf{u}(\mathbf{g})$ is ill-conditioned for vanished gradients ($\mathbf{g} = 0$) in early stage ($t$ is small).*

*Proof.* Considering that the $\mathbf{u}(\mathbf{g})$ is differentiable, the absolute condition number $\hat{\kappa}$ for $\mathbf{u}(\mathbf{g}_t)$ is

$$\hat{\kappa} = \lim_{\delta \to 0} \sup_{||\delta \mathbf{g}|| \le \delta} \frac{||\mathbf{u}(\mathbf{g} + \delta \mathbf{g}) - \mathbf{u}(\mathbf{g})||}{||\delta \mathbf{g}||} = ||\mathbf{J}(\mathbf{g})|| = \sqrt{\sum_{i=1}^{d}\left(\frac{\partial u}{\partial g_i}\right)^2}.$$

The full expression of $\frac{\partial u}{\partial g}$ can be found in Appendix B. In the early stage (i.e., $t$ is small), for the vanished gradient ($g_i = 0$), the absolute condition number $\hat{\kappa}$ is

$$\hat{\kappa} = \alpha \frac{1 - \beta_1}{1 - \beta_1^t} \sqrt{\sum_{i=1}^{d} \frac{1}{\epsilon + \sqrt{\frac{\beta_2 v_i^{(t-1)}}{1 - \beta_2^t}}}} \approx \frac{\alpha \sqrt{d}}{\epsilon}. \tag{3}$$

For example, in a classic setting where $d = 1024, \epsilon = 10^{-6}, \alpha = 10^{-4}$, we have $\hat{\kappa} = 3200$, which is a very large number. This tells us that in early stage, the $\mathbf{u}(\mathbf{g}_t)$ is ill-conditioned.

$\square$

Intuitively, when there is a small noise $||\delta \mathbf{g}|| \le \delta$ added to the gradient $\mathbf{g}$, the change of the update $||\mathbf{u}(\mathbf{g} + \delta \mathbf{g}) - \mathbf{u}(\mathbf{g})||$ could be thousand times larger than $||\delta \mathbf{g}||$. This will make the training

unstable and vulnerable to a small perturbation. This study is also consistent with the empirically findings by Wang et al. (2022a) that the exploding gradients in higher layers is not the root cause of Post-LN training difficultly. Further more, to verify our approximation, we also have simulation in Appendix B.

Moreover, from Equation (3), given a fixed model with width $d$, seems there are two possible way to reduce the $\hat{\kappa}$: increasing the $\epsilon$ or decreasing the $\alpha$. However, the first one is not viable because a large $\epsilon$ will make an adaptive optimizer less adaptive. Therefore, in practise, researchers have to reduce the learning-rate $\alpha$ (e.g., using learning-rate warm-up) to ease this problem.

To conclude, as the gradient vanishing is a critical issue even when the model is trained with adaptive optimizes. As a result, we purpose to solve this problem from the architecture aspect.

### 3.2 The Representation Collapse Issue

**The Representation Collapse in Pre-LN** The issue with the representation capability of Pre-LN was initially observed by Liu et al. (2020). In summary, the Pre-LN Transformer's hidden representation cannot be refined by deeper layers due to the normalization of layer outputs. In this work, we propose a novel analysis approach that directly examines the distribution of hidden state changes, represented by $|x_{k+1}^{ln} - x_k^{ln}|$, and output changes, denoted by $|y_N - y_{N-1}|$. Our new method offers a straightforward way to obtain quantitative results regarding the convergence rate.

**Theorem 3.3.** *For Pre-LN, assume* $x_k^f \sim \mathcal{N}(0, \sigma^2 I)$ *independently for all* $k \in [N]$, *we have* $x_{k+1}^{ln} - x_k^{ln} \sim \mathcal{N}(0, \omega_k^2) I$ *where* $\omega_k^2 = \frac{2}{\sqrt{k}(\sqrt{k-1}+\sqrt{k})}$.

*Proof.* As $x_k^f \sim \mathcal{N}(0, \sigma^2 I)$, we have $x_k^a = \sum_{j=1}^{k-1} x_j^f$ thus $x_k^a \sim \mathcal{N}(0, (k-1)\sigma^2 I)$. For the normalization layer, we approximate its effect as follows, $x_k^{ln} = \frac{x_k^a}{\sqrt{k-1}\sigma}$. Then we have

$$x_{k+1}^{ln} - x_k^{ln} = \frac{x_{k+1}^a}{\sqrt{k}\sigma} - \frac{x_k^a}{\sqrt{k-1}\sigma} = \frac{\sqrt{k-1} - \sqrt{k}}{\sqrt{k(k-1)}\sigma} \cdot x_k^a + \frac{1}{\sqrt{k}\sigma} \cdot x_k^f.$$

We know that $\frac{\sqrt{k-1}-\sqrt{k}}{\sqrt{k(k-1)}\sigma} \cdot x_k^a \sim \mathcal{N}(0, \frac{(\sqrt{k-1}-\sqrt{k})^2}{k} I)$ and $\frac{1}{\sqrt{k}\sigma} \cdot x_k^f \sim \mathcal{N}(0, \frac{1}{k} I)$. Because $x_k^a$ and $x_k^f$ are independent, we have $a_{k+1} - a_k \sim \mathcal{N}(0, \omega_k^2 I)$ and $\omega_k^2 = \frac{(\sqrt{k-1}-\sqrt{k})^2}{k} + \frac{1}{k} = \frac{2}{\sqrt{k}(\sqrt{k-1}+\sqrt{k})}$. $\square$

**Corollary 3.4.** *For each coordinate* $i$ *of* $x_{k+1}^{ln} - x_k^{ln}$, *we have* $\mathbb{E}[|(x_{k+1}^{ln} - x_k^{ln})_i|] \sim O(\frac{1}{\sqrt{k}})$

From Corollary 3.4, we can see that the expectation of $|(a_{k+1} - a_k)_i|$ decreases to 0 as $k$ increases to infinity with rate $1/\sqrt{k}$. This means, when the number of layers increases, the inputs to later layers will be similar to each other. Thus, the capability of the later layers are not fully used because they cannot further refine the representations.

**Corollary 3.5.** *When adding an extra layer to a* $N-1$ *layer Pre-LN Transformer, the output difference* $\mathbb{E}[|(y_N - y_{N-1})_i|] \sim O(\frac{1}{\sqrt{N}})$ *for each coordinate* $i$.

The proof of Corollary 3.5 is in Appendix C, it means that adding extra layer in the deep Pre-LN Transformer has little impact on the output. Intuitively, this means the extra layer also cannot refine the model outputs and the model's capacity is not fully used.

### 3.3 Analysis of ResiDual

**ResiDual Does Not Suffer From Gradient Vanishing Issue** For the ResiDual architecture (Figure 1c), we can view it as a mixture of Post-LN Transformer and Pre-LN Transformer. Specifically, in the forward process, ResiDual Transformer behaves exactly the same as Post-LN except adding a dual branch of normalized sum of all block outputs in the end. In the backward process, the error signal back-propagates through both branches. We can explicitly write down the gradients at block

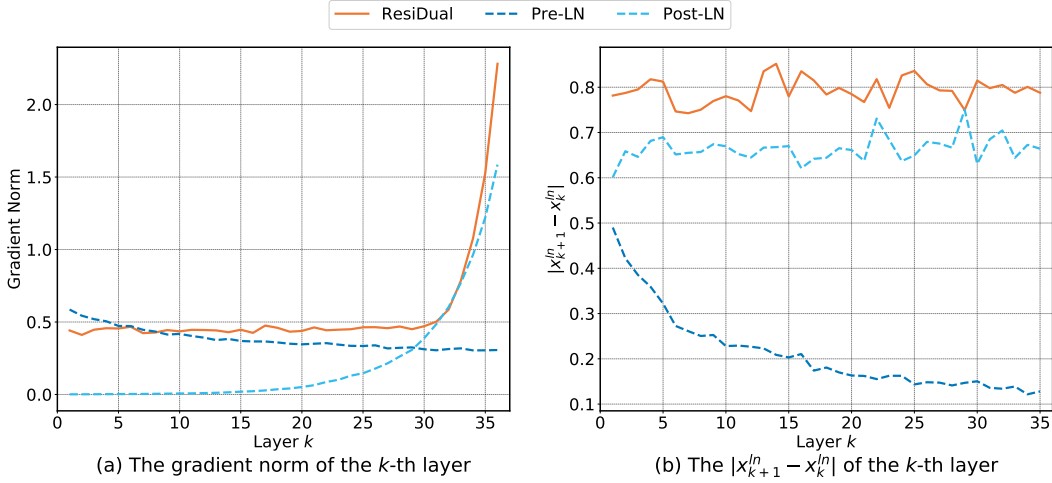

(a) The gradient norm of the $k$-th layer    (b) The $|x_{k+1}^{ln} - x_k^{ln}|$ of the $k$-th layer

Figure 2: Study of the Gradient norm and hidden representation w.r.t layer $k$ in each method. Details in Appendix E

$k$ as follows

$$\frac{\partial \mathcal{L}}{\partial \boldsymbol{w}_k} = \left(\frac{\partial \mathcal{L}}{\partial \boldsymbol{w}_k}\right)_{post} + \left(\frac{\partial \mathcal{L}}{\partial \boldsymbol{w}_k}\right)_{dual}, \tag{4}$$

where $\left(\frac{\partial \mathcal{L}}{\partial \boldsymbol{w}_k}\right)_{post}$ denotes the gradient component from the Post-LN branch and $\left(\frac{\partial \mathcal{L}}{\partial \boldsymbol{w}_k}\right)_{dual}$ denotes the gradient component from the dual branch. Specifically,

$$\left(\frac{\partial \mathcal{L}}{\partial \boldsymbol{w}_k}\right)_{post} = \frac{\partial \mathcal{L}}{\partial \overrightarrow{\boldsymbol{x}}_{N+1}} \left(\prod_{l=k}^{N} \frac{\partial \overrightarrow{\boldsymbol{x}}_{l+1}}{\partial \overrightarrow{\boldsymbol{x}}_l^{ln}} \frac{\partial \overrightarrow{\boldsymbol{x}}_l^{ln}}{\partial \overrightarrow{\boldsymbol{x}}_l}\right) \frac{\partial \overrightarrow{\boldsymbol{x}}_k^f}{\partial \boldsymbol{w}_k} = \frac{\partial \mathcal{L}}{\partial \overrightarrow{\boldsymbol{x}}_{N+1}} \left(\prod_{l=k}^{N} \left(\boldsymbol{I} + \frac{\partial \overrightarrow{\boldsymbol{x}}_l^f}{\partial \overrightarrow{\boldsymbol{x}}_l^{ln}}\right) \frac{\partial \overrightarrow{\boldsymbol{x}}_l^{ln}}{\partial \overrightarrow{\boldsymbol{x}}_l}\right) \frac{\partial \overrightarrow{\boldsymbol{x}}_k}{\partial \boldsymbol{w}_k},$$

and

$$\left(\frac{\partial \mathcal{L}}{\partial \boldsymbol{w}_k}\right)_{dual} = \frac{\partial \mathcal{L}}{\partial \overrightarrow{\boldsymbol{x}}_{N+1}} \left(\prod_{l=k+1}^{N} \frac{\partial \overrightarrow{\boldsymbol{x}}_{l+1}}{\partial \overrightarrow{\boldsymbol{x}}_l}\right) \frac{\partial \overrightarrow{\boldsymbol{x}}_{k+1}^f}{\partial \boldsymbol{w}_k} = \frac{\partial \mathcal{L}}{\partial \overrightarrow{\boldsymbol{x}}_{N+1}} \left(\prod_{l=k+1}^{N} \left(\boldsymbol{I} + \frac{\partial \overrightarrow{\boldsymbol{x}}_l^f}{\partial \overrightarrow{\boldsymbol{x}}_l^{ln}} \frac{\partial \overrightarrow{\boldsymbol{x}}_l^{ln}}{\partial \overrightarrow{\boldsymbol{x}}_l}\right)\right) \frac{\partial \overrightarrow{\boldsymbol{x}}_k^f}{\partial \boldsymbol{w}_k}.$$

We see that when $k$ is small, the Pre-LN gradient component dominates and when $k$ is close to $N$, the Post-LN gradient component dominates. It is safe to estimate the gradient norm of the $k$-th block in ResiDual Transformer as follows,

$$\left\|\frac{\partial \mathcal{L}}{\partial \boldsymbol{w}_k}\right\|_F \approx \max \left\{\mathcal{O}\left((1/2)^{(N-k)/2} e^{\sqrt{N-k}}\right), \mathcal{O}\left(\sqrt{\frac{\log(N-k)\cdot}{N}}\right)\right\}, \tag{5}$$

where again we ignore the terms irrelevant with $N, k$. Therefore, the ResiDual architecture does not suffer gradient vanishing problem. It is worthy to note gradient vanishing problem does not directly relate to inefficient training because in Adam the actual update is rescaled to be normal even if extreme small gradient is obtained. However, the gradient vanishing problem would affect the stability of the Adam optimizer as we argue as follows.

In Figure 2(a), we show the gradient distribution for different methods. We can find that the Post-LN has almost zero gradient for early layers, while the ResiDual (orange line) do not have such an issue. The clearly shows that our method can ensure a lower-bound of the gradient norm. Meanwhile, note that non of these models have the exploding-gradient issue. According to Theorem 3.1, the gradient of last layer (i.e., $k = N$) is not related to $N$.

**ResiDual Does Not Suffer From Representation Collapse Issue.**    The Post-LN and ResiDual do not have the representation collapse issue. Formally,

**Theorem 3.6.** *In Post-LN and ResiDual, assume $\boldsymbol{x}_k^f \sim \mathcal{N}(0, \sigma^2 \boldsymbol{I})$ independently for all $k \in [N]$, the $\boldsymbol{x}_{k+1}^{ln} - \boldsymbol{x}_k^{ln} \sim \mathcal{N}(0, \omega^2)$ where $\omega$ is not related to $k$.*

*Proof.* As $\boldsymbol{x}_{k+1}^{ln} = \text{LN}(\boldsymbol{x}_k^a) = \text{LN}(\boldsymbol{x}_k^{ln} + \boldsymbol{x}_k^f)$, and $\boldsymbol{x}_k^{ln} \sim \mathcal{N}(0, \boldsymbol{I}), \boldsymbol{x}_k^f \sim \mathcal{N}(0, \sigma^2 \boldsymbol{I})$, we have

$$\boldsymbol{x}_{k+1}^{ln} - \boldsymbol{x}_k^{ln} = \frac{\boldsymbol{x}_k^{ln} + \boldsymbol{x}_k^f}{\sqrt{1 + \sigma^2}} - \boldsymbol{x}_k^{ln} = \frac{(1 - \sqrt{1 + \sigma^2})\boldsymbol{x}_k^{ln} + \boldsymbol{x}_k^f}{\sqrt{1 + \sigma^2}}.$$

Thus, $\boldsymbol{x}_{k+1}^{ln} - \boldsymbol{x}_k^{ln} \sim \mathcal{N}(0, \omega^2)$ where $\omega^2 = 2 - 2\frac{\sqrt{1+\sigma^2}}{1+\sigma^2}$ and $\omega$ is not related to $k$. □

**Corollary 3.7.** *When adding an extra layer to a $N-1$ layer Pre-LN Transformer, the output difference $\mathbb{E}[|(\boldsymbol{y}_N - \boldsymbol{y}_{N-1})_i|] \geq \sqrt{\frac{2}{\pi}}\omega$ for each coordinate $i$.*

The proof of 3.7 is in the supplementary material. From these analyse, we can see that the variance of $\boldsymbol{x}_{k+1}^{ln} - \boldsymbol{x}_k^{ln}$ will not decrease when the depth increases, so that later layers can continue refining the hidden representation. Meanwhile, according to Corollary 3.7, the model output can also be refined with a lower bound that not related to depth. In another words, ResiDual can avoid the representation bottleneck of Pre-LN model. To demonstrate this, we also show the $|\boldsymbol{x}_{k+1}^{ln} - \boldsymbol{x}_k^{ln}|$ for different architectures in Figure 2(b). As the lines show, our method (orange line) has a consistent value of $|\boldsymbol{x}_{k+1}^{ln} - \boldsymbol{x}_k^{ln}|$, while the Pre-LN's value will decrease when the depth is high.

## 4 EXPERIMENTS

### 4.1 EXPERIMENTAL SETTINGS

**Data** We conducted experiments on three datasets: the IWSLT-14 English to German (EN→DE) dataset (Cettolo et al., 2014), the WMT German to English (DE→EN) dataset (Bojar et al., 2014), and the OPUS-100 multilingual dataset (Zhang et al., 2020). More details are in Appendix H.

**Model** Our model is implemented using the FairSeq (Ott et al., 2019) framework with conventional settings as previous works. Notably, our method introduce only negligible parameters to the vanilla Transformer network. Meanwhile, given that the residual connection operations have a relatively small computational cost compared to Attention and FFN layers, the efficiency of our method should not hinder its practical use. We empirically observed about 3% increase in computation cost. Please refer to the Appendix H for hyper-parameters.

### 4.2 EXPERIMENTAL RESULTS ON IWSLT

The experimental results of the IWSLT'14 dataset are presented in Table 2. Two types of models were used: shallow models with 6-layer encoders and 6-layer decoders (E6D6), and deep models with 12-layer encoders and 12-layer decoders (E12D12). We made the following observations:

Firstly, the Post-LN method was successful in converging for E6D6 but not for E12D12. Secondly, the Pre-LN method converged in both depths, but its performance (35.12, 35.18) was inferior to that of the Post-LN E6D6 (35.37) or our E6D6 (35.63). Thirdly, the methods such as DeepNet (Wang et al., 2022a) and Admin (Liu et al., 2020) only showed a slight improvement over the vanilla models, and our method

Table 2: Experimental Results on IWSLT.

| Method | E6D6 | E12D12 |
|---|---|---|
| Post-LN | 35.37 | Fail |
| Pre-LN | 35.12 | 35.18 |
| DeepNet | 35.34 | 35.39 |
| Admin | 35.50 | 35.67 |
| T-Fixup | 34.88 | 35.45 |
| NormFormer | 35.14 | 31.00 |
| **ResiDual(Ours)** | **35.63** | **36.09** |

achieved best performance. Especially, in E12D12, we have 0.9-point BLEU gain over the standard Pre-LN model. Our preliminary experiments revealed that increasing the model depth further led to over-fitting issues for all models due to limited data. Therefore, we do not report 18 layer model results on this dataset.

### 4.3 EXPERIMENTAL RESULTS ON WMT

The experimental results on shallow (E6D6) and deep (E18D18) models are presented in Table 3. We only report the average score here and more details can be found in Table 6 and Table 7 in Appendix F. Firstly, we find that the Post-LN model can only converge in the E6D6 setting but not in E18D18 setting. Secondly, the Pre-LN model shows convergence in both E6D6 and E18D18. However, the performance of the Pre-LN model in E18D18 (26.57) is similar to Post-LN model in E6D6 (26.59). Finally, our method achieved the best performance for both shallow and deep models. Particularly, we observed an improvement over the Pre-LN performance by 1.1-point for the E18D18 model.

Table 3: Experimental Results on WMT.

| Method | E6D6 | E18D18 |
|---|---|---|
| Pre-LN | 26.10 | 26.57 |
| Post-LN | 26.59 | Fail |
| DLCL | 26.52 | 26.90 |
| T-Fixup | 26.43 | 26.94 |
| DeepNet | 26.38 | 27.13 |
| Admin | 26.49 | 26.86 |
| B2T | 26.53 | 27.30 |
| **ResiDual** | **26.85** | **27.65** |

### 4.4 EXPERIMENTAL RESULTS ON OPUS-100

We evaluate our method on the OPUS-100 dataset, which consists of 100 language pairs and $55M$ parallel sentence pairs. Because we trained single model for both from English (EX) and to English (XE) direction, the total data size is about $110M$ sentence pairs and approximately 4 billion tokens. Table 4 shows the experimental results. In addition to the original baselines provided by Zhang et al. (2020), we also reproduced the 18-layer encoder and 18-layer decoder model (E18D18). We found that the Post-LN model failed to converge thus only show the Pre-LN results in Table 4. As we can see from the table,

Table 4: Experimental Results on OPUS-100. *Denotes The Results From Zhang et al. (2020).

| Method | #Layers | EX | XE | ALL |
|---|---|---|---|---|
| | 6 | 21.4 | 27.5 | 24.5 |
| Pre-LN* | 12 | 22.9 | 29.5 | 26.2 |
| | 24 | 24.0 | 31.4 | 27.7 |
| Pre-LN | 18 | 27.9 | 32.8 | 30.3 |
| DeepNet | 100 | 29.0 | 33.2 | 31.1 |
| **ResiDual** | 18 | 28.7 | 33.4 | 31.0 |

our method achieves about 0.7 BLEU points over the standard Pre-LN model. The BLEU score is almost identical to a 100-layer DeepNet (Wang et al., 2022a) model, which is about 5 times deeper of our model. This demonstrates that our model can more effectively use deeper layers.

### 4.5 STUDY OF LEARNING-RATE WARM-UP

One of the objectives of our approach is to facilitate easy and stable training for Transformer models. Therefore, we conducted experiments using different learning rate schedules on the IWSLT dataset. Table 5 presents the results for various models with or without learning-rate warm-up. Further details can be found in the Appendix G. We observe that Post-LN necessitates warm-up for convergence, while Pre-LN and our method are not. This is consistent with our study in Section 3.

Table 5: Study on the Use of Learning-Rate Warm-Up.

| Method | Post-LN | | Pre-LN | | ResiDual | |
|---|---|---|---|---|---|---|
| LR Warm-Up | Yes | No | Yes | No | Yes | No |
| E6D6 | 35.37 | Fail | 35.12 | 32.28 | 35.63 | **35.76** |
| E12D12 | Fail | Fail | 35.18 | 31.82 | **36.09** | 35.57 |

## 5 CONCLUSION

This research is to advance the Transformer architecture and offers an effective strategy for optimizing it with enhanced performance. This paper first examines the limitations of two widely employed variants, and introduces a novel approach, referred to as ResiDual, to mitigate both issues. ResiDual consists two residual connections to circumvent the gradient vanishing and the representation collapse problem. Theoretical analysis and empirical results validates that the suggested model can surmount both challenges while preserving the advantages of each residual connection.

## REPRODUCIBILITY STATEMENT

The complete proof can be found in Appendix A, B, C, and D. The detailed process to build Figure 2 is in Appendix E. Our code is anonymously available at `https://anonymous.4open.science/r/residual_review-6F08`. Meanwhile, you can refer Appendix H for implementation details like data processing scripts and hyper parameters.

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

## A  PROOF OF THEOREM 3.1

*Proof.* For the Post-LN Transformer, the gradient of the parameters in the $k$-th layer (take $\boldsymbol{W}_k$ as an example) can be written as

$$\frac{\partial \mathcal{L}}{\partial \boldsymbol{w}_k} = \frac{\partial \mathcal{L}}{\partial \overrightarrow{\boldsymbol{x}}_{N+1}^{ln}} \frac{\partial \overrightarrow{\boldsymbol{x}}_{N+1}^{ln}}{\partial \overrightarrow{\boldsymbol{x}}_N^a} \left( \prod_{l=k}^{N-1} \frac{\partial \overrightarrow{\boldsymbol{x}}_{l+1}^a}{\partial \overrightarrow{\boldsymbol{x}}_{l+1}^{ln}} \frac{\partial \overrightarrow{\boldsymbol{x}}_{l+1}^{ln}}{\partial \overrightarrow{\boldsymbol{x}}_l^a} \right) \frac{\partial \overrightarrow{\boldsymbol{x}}_k^f}{\partial \boldsymbol{w}_k}$$

$$= \frac{\partial \mathcal{L}}{\partial \overrightarrow{\boldsymbol{x}}_{N+1}^{ln}} \frac{\partial \overrightarrow{\boldsymbol{x}}_{N+1}^{ln}}{\partial \overrightarrow{\boldsymbol{x}}_N^a} \left( \prod_{l=k}^{N-1} \left( \boldsymbol{I} + \frac{\partial \overrightarrow{\boldsymbol{x}}_{l+1}^f}{\partial \overrightarrow{\boldsymbol{x}}_{l+1}^{ln}} \right) \frac{\partial \overrightarrow{\boldsymbol{x}}_{l+1}^{ln}}{\partial \overrightarrow{\boldsymbol{x}}_l^a} \right) \frac{\partial \overrightarrow{\boldsymbol{x}}_k^f}{\partial \boldsymbol{w}_k}.$$

We care about the spectral norm of the term $\frac{\partial \overrightarrow{\boldsymbol{x}}_{N+1}^{ln}}{\partial \overrightarrow{\boldsymbol{x}}_N^a} \left( \prod_{l=k}^{N-1} \left( \boldsymbol{I} + \frac{\partial \overrightarrow{\boldsymbol{x}}_{l+1}^f}{\partial \overrightarrow{\boldsymbol{x}}_{l+1}^{ln}} \right) \frac{\partial \overrightarrow{\boldsymbol{x}}_{l+1}^{ln}}{\partial \overrightarrow{\boldsymbol{x}}_l^a} \right)$, which varies for different blocks.

For the feedforward layer and attention layer, we respectively have $l \in [N]$,

$$\frac{\partial \overrightarrow{\boldsymbol{x}}_l^f}{\partial \overrightarrow{\boldsymbol{x}}_l^{ln}} = \begin{pmatrix} \boldsymbol{W}_l^T & & \\ & \ddots & \\ & & \boldsymbol{W}_l^T \end{pmatrix} \quad \text{and} \quad \frac{\partial \overrightarrow{\boldsymbol{x}}_l^f}{\partial \overrightarrow{\boldsymbol{x}}_l^{ln}} = \begin{pmatrix} \frac{1}{n}\boldsymbol{W}_{V,l}^T & \cdots & \frac{1}{n}\boldsymbol{W}_{V,l}^T \\ \vdots & \ddots & \vdots \\ \frac{1}{n}\boldsymbol{W}_{V,l}^T & \cdots & \frac{1}{n}\boldsymbol{W}_{V,l}^T \end{pmatrix},$$

based on the setup of the feedforward layer and attention layer at the initialization. For the layer normalization layer, we have

$$\frac{\partial \overrightarrow{\boldsymbol{x}}_{l+1}^{ln}}{\partial \overrightarrow{\boldsymbol{x}}_l^a} = \begin{pmatrix} \frac{\partial \text{LN}(\boldsymbol{x}_l^{a(1)})}{\partial \boldsymbol{x}_l^{a(1)}} & & \\ & \ddots & \\ & & \frac{\partial \text{LN}(\boldsymbol{x}_l^{a(n)})}{\partial \boldsymbol{x}_l^{a(n)}} \end{pmatrix} = \begin{pmatrix} \frac{\sqrt{d}}{\|\boldsymbol{x}_l^{a(1)}\|_2}\boldsymbol{I} & & \\ & \ddots & \\ & & \frac{\sqrt{d}}{\|\boldsymbol{x}_l^{a(n)}\|_2}\boldsymbol{I} \end{pmatrix},$$

as we assume on the Jacobian of layer normalization.

We note that $\boldsymbol{I} + \frac{\partial \overrightarrow{\boldsymbol{x}}_l^f}{\partial \overrightarrow{\boldsymbol{x}}_l^{ln}}$ are block-circulant matrices for all $l$ and the product of block-circulant matrices is also block-circulant. We know a block-circulant matrix has the following property

$$\left\| \begin{matrix} \boldsymbol{B} & \boldsymbol{A} & \cdots & \boldsymbol{A} \\ \boldsymbol{A} & \boldsymbol{B} & \cdots & \boldsymbol{A} \\ \vdots & & \ddots & \vdots \\ \boldsymbol{A} & \cdots & \boldsymbol{A} & \boldsymbol{B} \end{matrix} \right\|_2 = \|\boldsymbol{B} + (n-1)\boldsymbol{A}\|_2,$$

where $\boldsymbol{B}$ and $\boldsymbol{A}$ are square matrices and there are $n-1$ $\boldsymbol{A}$s each row. Hence we have

$$\left\| \frac{\partial \overrightarrow{\boldsymbol{x}}_{N+1}^{ln}}{\partial \overrightarrow{\boldsymbol{x}}_N^a} \left( \prod_{l=k}^{N-1} \left( \boldsymbol{I} + \frac{\partial \overrightarrow{\boldsymbol{x}}_{l+1}^f}{\partial \overrightarrow{\boldsymbol{x}}_{l+1}^{ln}} \right) \frac{\partial \overrightarrow{\boldsymbol{x}}_{l+1}^{ln}}{\partial \overrightarrow{\boldsymbol{x}}_l^a} \right) \right\|_2 = \left( \prod_{l=k}^{N} \frac{\sqrt{d}}{\|\boldsymbol{x}_l^{a(i)}\|_2} \right) \left\| \left( \prod_{l=k+1}^{N} (\boldsymbol{I} + \boldsymbol{w}_l^T) \right) \right\|_2,$$

where $\boldsymbol{w}_l$ represents either $\boldsymbol{W}_{V,l}$ or $\boldsymbol{W}_l$. We know that with high probability, $\|\boldsymbol{x}_l^{a(i)}\|_2 \in (1\pm\epsilon)\sqrt{2d}$ where $\epsilon$ is a small positive constant, based on the assumption $\|\boldsymbol{x}_l^{ln(i)}\|_2 = \sqrt{d}$ and the random initialization of $\boldsymbol{w}_l$ for all $i \in [n]$. Thus we have a term $\left(\prod_{l=k}^{N} \frac{\sqrt{d}}{\|\boldsymbol{x}_l^{a(i)}\|_2}\right) \approx \mathcal{O}\left((1/2)^{(N-k)/2}\right)$. Moreover, based on the random matrix argument Zhang et al. (2022), we have with high probability,

$$\left\|\prod_{l=k}^{N}(\boldsymbol{I} + \boldsymbol{w}_l^T)\right\|_2 \approx \mathcal{O}(e^{\sqrt{N-k}}).$$

Therefore, we have $\|\frac{\partial \mathcal{L}}{\partial \boldsymbol{w}_k}\|_F \approx \mathcal{O}((1/2)^{(N-k)/2} \cdot e^{\sqrt{N-k}})$, which diminishes exponentially as $N-k$ is large.

On the other hand, we have the bound for Pre-LN transformer as follows.

$$\frac{\partial \mathcal{L}}{\partial \boldsymbol{w}_k} = \frac{\partial \mathcal{L}}{\partial \boldsymbol{y}}\frac{\partial \boldsymbol{y}}{\partial \overrightarrow{\boldsymbol{x}}_{N+1}^a}\left(\prod_{l=k+1}^{N} \frac{\partial \overrightarrow{\boldsymbol{x}}_{l+1}^a}{\partial \overrightarrow{\boldsymbol{x}}_l^a}\right)\frac{\partial \overrightarrow{\boldsymbol{x}}_k^f}{\partial \boldsymbol{w}_k} = \frac{\partial \mathcal{L}}{\partial \boldsymbol{y}}\frac{\partial \boldsymbol{y}}{\partial \overrightarrow{\boldsymbol{x}}_{N+1}^a}\left(\prod_{l=k+1}^{N}\left(\boldsymbol{I} + \frac{\partial \overrightarrow{\boldsymbol{x}}_l^f}{\partial \overrightarrow{\boldsymbol{x}}_l^{ln}}\frac{\partial \overrightarrow{\boldsymbol{x}}_l^{ln}}{\partial \overrightarrow{\boldsymbol{x}}_l^a}\right)\right)\frac{\partial \overrightarrow{\boldsymbol{x}}_k^f}{\partial \boldsymbol{w}_k}.$$

We know that with high probability, $\|\boldsymbol{x}_l^{a(i)}\|_2 \in (1\pm\epsilon)\sqrt{ld}$ based on the assumption $\|\boldsymbol{x}_l^{ln(i)}\|_2 = \sqrt{d}$ and the random initialization of $\boldsymbol{w}_l$ for all $i \in [n]$. Hence $\frac{\partial \boldsymbol{x}_l^{ln}}{\partial \boldsymbol{x}_l^a} \approx 1/\sqrt{l}\boldsymbol{I}$. Therefore, with high probability, we have

$$\left\|\prod_{l=k+1}^{N}\left(\boldsymbol{I} + \frac{\partial \overrightarrow{\boldsymbol{x}}_l^f}{\partial \overrightarrow{\boldsymbol{x}}_l^{ln}}\frac{\partial \overrightarrow{\boldsymbol{x}}_l^{ln}}{\partial \overrightarrow{\boldsymbol{x}}_l^a}\right)\right\|_2 \approx \left\|\prod_{l=k+1}^{N}(\boldsymbol{I} + \frac{1}{\sqrt{l}}\boldsymbol{w}_l^T)\right\|_2 = \mathcal{O}(\log(N-k)),$$

where the last inequality is based on the argument for the product of random matrices (Zhang et al., 2022). Therefore, by further being aware of $\|\frac{\partial \boldsymbol{y}}{\partial \overrightarrow{\boldsymbol{x}}_{N+1}^a}\|_2 \approx 1/\sqrt{N}$, we have $\|\frac{\partial \mathcal{L}}{\partial \boldsymbol{w}_k}\| \approx \mathcal{O}(\frac{\log(N-k)}{\sqrt{N}})$, which scales with $\sqrt{N}$ inverse proportionally.

$\square$

# B  STUDY OF ADAM

The Adam update formula is

$$\mathbf{w}^{(t)} \leftarrow \mathbf{w}^{(t-1)} - \alpha \cdot \hat{\mathbf{m}}^{(t)}/(\sqrt{\hat{\mathbf{v}}^{(t)}} + \epsilon)$$
$$\hat{\mathbf{m}}^{(t)} \leftarrow \mathbf{m}^{(t)}/(1-\beta_1^t), \quad \hat{\mathbf{v}}^{(t)} \leftarrow \mathbf{v}^{(t)}/(1-\beta_2^t)$$
$$\mathbf{m}^{(t)} \leftarrow \beta_1 \cdot \mathbf{m}^{(t-1)} + (1-\beta_1) \cdot \mathbf{g}$$
$$\mathbf{v}^{(t)} \leftarrow \beta_2 \cdot \mathbf{v}^{(t-1)} + (1-\beta_2) \cdot \mathbf{g}^2,$$

The full expression of $u'(g_{t,i})$ is

$$\frac{\partial u}{\partial g} = \frac{\alpha g \sqrt{\frac{\beta_2 v^{(t-1)} + g^2 \cdot (1-\beta_2)}{1-\beta_2^t}}(1-\beta_2)\left(\beta_1 m^{(t-1)} + g(1-\beta_1)\right)}{(1-\beta_1^t)\left(\epsilon + \sqrt{\frac{\beta_2 v^{(t-1)} + g^2 \cdot (1-\beta_2)}{1-\beta_2^t}}\right)^2\left(\beta_2 v^{(t-1)} + g^2 \cdot (1-\beta_2)\right)}$$
$$+ \frac{\alpha(1-\beta_1)}{(1-\beta_1^t)\left(\epsilon + \sqrt{\frac{\beta_2 v^{(t-1)} + g^2 \cdot (1-\beta_2)}{1-\beta_2^t}}\right)} \tag{6}$$

When the gradient $g = 0$, we have

$$\frac{\partial u}{\partial g} = \frac{\alpha(1-\beta_1)}{(1-\beta_1^t)\left(\epsilon + \sqrt{\frac{\beta_2 v_{t-1,i}}{1-\beta_2^t}}\right)}$$

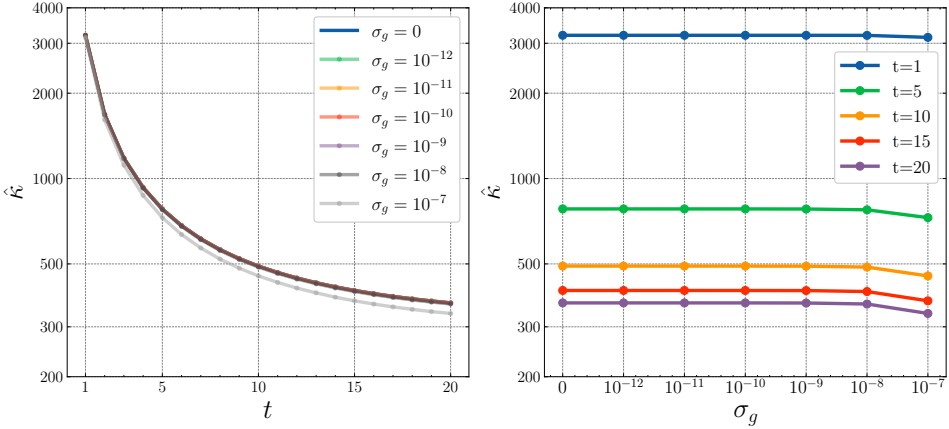

Figure 3: The absolute condition number $\hat{\kappa}$ w.r.t $t$ (left) and $\sigma_g$ (right).

To simulate the Adam update and compute the $\hat{\kappa}$, we use the parameter as $d = 1024, \epsilon = 10^{-6}, \beta_1 = 0.9, \beta_2 = 0.98, \alpha = 10^{-4}$. Then for each step, we random first sample $\mathbf{g} \sim \mathcal{N}(0, \sigma_g^2 \mathbf{I})$, can compute the $\hat{\kappa}$ based on full Equation (6). Finally, we update the Adam momentum with its update rules. In Figure 3, we show the simulated results by sampling $\mathbf{g} \sim \mathcal{N}(0, \sigma_g^2 \mathbf{I})$ where $\sigma_g$ ranges from 0 to $10^{-7}$. In the left plot, we show how $\hat{\kappa}$ change w.r.t $t$ for different $\sigma_g$. We can find that our estimation of $\hat{\kappa}$ is accurate as most of lines are overlaped. Besides, it also show that even after 20 steps update, the $\hat{\kappa}$ is still greater than 300. As many lines overlapped in the left plot, on the right side, we show a zoomed-in view by selecting five timestamp ans show the $\hat{\kappa}$ w.r.t to $\sigma_g$. It is clear that $\hat{\kappa}$ is large when $\sigma_g$ is small.

## C    PROOF OF COROLLARY 3.5

Given two Pre-LN Transformer with $N - 1$ and $N$ layers, we denote their output as $\boldsymbol{y}_{N-1}$ and $\boldsymbol{y}_N$, respectively. Then we have

$$\boldsymbol{y}_{N-1} = \mathrm{LN}(\boldsymbol{x}_{N-1}^a + \boldsymbol{x}_{N-1}^f) = \boldsymbol{x}_N^{ln}$$
$$\boldsymbol{y}_N = \mathrm{LN}(\boldsymbol{x}_N^a + \boldsymbol{x}_N^f) = \boldsymbol{x}_{N+1}^{ln}$$
$$|\boldsymbol{y}_N - \boldsymbol{y}_{N-1}| = |\boldsymbol{x}_{N+1}^{ln} - \boldsymbol{x}_N^{ln}|$$

From Corollary 3.4, we can approve that $\mathbb{E}[|(\boldsymbol{y}_N - \boldsymbol{y}_{N-1})_i|] \sim O(\frac{1}{\sqrt{N}})$.

## D    PROOF OF COROLLARY 3.7

*Proof.*

$$\boldsymbol{y}_N - \boldsymbol{y}_{N-1} = \left(\boldsymbol{x}_N^{ln} + \mathrm{LN}\left(\mathbf{x}_{N+1}^d\right)\right) - \left(\boldsymbol{x}_{N-1}^{ln} + \mathrm{LN}\left(\mathbf{x}_N^d\right)\right)$$
$$= \left(\boldsymbol{x}_N^{ln} - \boldsymbol{x}_{N-1}^{ln}\right) + \left(\mathrm{LN}\left(\mathbf{x}_{N+1}^d\right) - \mathrm{LN}\left(\mathbf{x}_N^d\right)\right)$$

The $\mathrm{LN}\left(\mathbf{x}_{N+1}^d\right) - \mathrm{LN}\left(\mathbf{x}_N^d\right)$ is also a zero-mean Gaussian distribution, which can be denoted as $\mathcal{N}(0, \hat{\omega}_N^2)$. Then we have $\boldsymbol{y}_N - \boldsymbol{y}_{N-1} \sim \mathcal{N}(0, \omega^2 + \hat{\omega}_N^2)$. Therefore,

$$\mathbb{E}[|(\boldsymbol{y}_N - \boldsymbol{y}_{N-1})_i|] = \sqrt{\frac{2}{\pi}}\sqrt{\omega^2 + \hat{\omega}_N^2}$$

$$\geq \sqrt{\frac{2}{\pi}}\omega$$

$\square$

# E  How to get the Figure 2

We created Transformer networks with 36 layers using different architectures: Pre-LN, Post-LN, and ResiDual. The embedding dimension, FFN dimension, and the number of attention heads were set to 256, 1024, and 4, respectively. We used the fairseq framework without modifying any initialization settings. We then created a dummy mini-batch with 16 sentences and 20 tokens per sentence. Next, we conducted a forward and backward process to collect the gradient norm and $x_k^{\text{ln}}$ of each layer. We hope this clarifies how we generated Figure 2, and we are happy to provide more details if needed.

# F  Full Results on WMT Dataset

The full results on WMT dataset is in Table 6 and Table 7. The baseline results are cited from Takase et al. (2022).

| Method | 2010 | 2011 | 2012 | 2013 | 2014 | 2015 | 2016 | Average |
|--------|------|------|------|------|------|------|------|---------|
| Pre-LN | 24.03 | 21.77 | 22.08 | 25.63 | 26.27 | 29.07 | 33.84 | 26.10 |
| Post-LN | 24.27 | 22.06 | 22.43 | 26.11 | 27.13 | 29.70 | 34.40 | 26.59 |
| DLCL | 23.94 | 22.00 | 22.24 | 26.11 | 27.37 | 29.71 | 34.26 | 26.52 |
| T-Fixup | 24.09 | 21.98 | 22.04 | 25.96 | 26.92 | 29.45 | 34.56 | 26.43 |
| DeepNet | 24.08 | 21.76 | 22.09 | 25.90 | 26.85 | 29.62 | 34.39 | 26.38 |
| Admin | 24.32 | 21.79 | 22.17 | 26.26 | 27.14 | 29.61 | 34.12 | 26.49 |
| B2T | 24.12 | 21.93 | 22.29 | 26.31 | 26.84 | 29.48 | **34.73** | 26.53 |
| **ResiDual(Ours)** | **24.42** | **22.20** | **22.66** | **26.64** | **27.23** | **30.22** | 34.55 | **26.85** |

Table 6: Experimental Results on WMT with E6D6 models.

| Method | 2010 | 2011 | 2012 | 2013 | 2014 | 2015 | 2016 | Average |
|--------|------|------|------|------|------|------|------|---------|
| Pre-LN | 24.07 | 21.98 | 22.4 | 26.28 | 27.36 | 29.74 | 34.16 | 26.57 |
| Post-LN | | | | Fail | | | | |
| DLCL | 24.20 | 22.51 | 22.83 | 26.59 | 27.97 | 30.24 | 33.98 | 26.90 |
| T-Fixup | 24.45 | 22.29 | 22.76 | 26.57 | 27.71 | 30.13 | 34.69 | 26.94 |
| DeepNet | 24.70 | 22.40 | 22.92 | 26.85 | 28.21 | 30.60 | 34.25 | 27.13 |
| Admin | 24.56 | 22.17 | 22.62 | 26.48 | 27.99 | 30.35 | 33.88 | 26.86 |
| B2T | 24.62 | 22.51 | 22.86 | 26.74 | 28.48 | 30.99 | 34.93 | 27.30 |
| **ResiDual(Ours)** | **24.85** | **22.76** | **23.18** | **27.60** | **28.79** | **31.12** | **35.24** | **27.65** |

Table 7: Experimental Results on WMT E18D18 models.

# G  Full Results on Learning-Rate Warm-Up

The full results on learning-rate warm-up is in Table 8.

# H  Implementation Details

## H.1  Data processing

The IWSLT-14 EN→DE dataset is relatively small, with only $140k$ sentence pairs. We followed the scripts in FairSeq (Ott et al., 2019) to preprocess the data. The WMT DE→EN dataset is larger, with $4M$ sentence pairs. We followed the preprocessing steps outlined in Takase & Kiyono (2021) by tokenizing the data with Moses tokenizer and then processing it with BPE (Sennrich et al., 2016). The model was trained on the WMT-14 training set and evaluated on the test set from years 2010 to 2016, following Takase & Kiyono (2021). The OPUS-100 dataset is a large-scale multilingual dataset containing 100 languages and approximately $55M$ sentence pairs. We used the script from Zhang et al.

| Method | Learing-Rate Scheduler | | E6D6 | E12D12 |
|---|---|---|---|---|
| | Warm-up | Decay Formula | | |
| Post-LN | Yes | Inverse Square Root | 35.37 | Fail |
| Post-LN | No | Inverse Square Root | Fail | Fail |
| Post-LN | No | Linear | Fail | Fail |
| Pre-LN | Yes | Inverse Square Root | 35.12 | 35.18 |
| Pre-LN | No | Inverse Square Root | 32.28 | 31.82 |
| Pre-LN | No | Linear | 32.26 | 31.85 |
| ResiDual(Ours) | Yes | Inverse Square Root | 35.63 | **36.09** |
| ResiDual(Ours) | No | Inverse Square Root | 35.76 | 35.57 |
| ResiDual(Ours) | No | Linear | **35.96** | 35.72 |

Table 8: Experimental Results on IWSLT with different learning-rate scheduler.

(2020) to tokenize the data and used SentencePiece (Kudo & Richardson, 2018) to segment the tokens. All data processing scripts are available in the Appendix. Because we train our model for both to and from English directions, the total training data is about $110M$.

The data processing scripts are

- IWSLT: `https://github.com/facebookresearch/fairseq/blob/main/examples/translation/prepare-iwslt14.sh`
- WMT: `https://github.com/facebookresearch/fairseq/blob/main/examples/translation/prepare-wmt14en2de.sh`
- OPUS-100: `https://github.com/bzhangGo/zero`

## H.2 HYPER-PARAMETERS

The training hyper-parameters are in Table 9, 10, and 11.

| Parameter | Value |
|---|---|
| Dropout | 0.3 |
| Embedding dim | 256 |
| FFN dim | 1024 |
| Attention heads | 4 |
| Encoder layers | 6/12 |
| Decoder layers | 6/12 |
| Learning rate | $5 * 10^{-4}$ |
| Learning rate scheduler | inverse sqrt |
| Warm-up steps | 4000 |
| Label smoothing | 0.1 |
| Weight decay | 0.0001 |
| Gradient clipping | 0 |
| Adam $\beta$ | 0.9, 0.98 |
| Max update steps | 300k |

Table 9: Hyper-parameters of IWSLT training

## H.3 IMPLEMENTATION TRICK ON FP16 TRAINING

In ResiDual, sometimes the $\mathbf{x}_k^d$ will exceed the value range that can be expressed by FP16 and may cause training error. When this happens, a simple numeric trick is to downscale $\mathbf{x}_k^d$ to make is within the FP16 scope. This will not affect the results because $\text{LN}(\mathbf{x}_{N+1}^d) = \text{LN}(\eta \cdot \mathbf{x}_{N+1}^d)$ for any $\eta > 0$. We did not observe such an issue in FP32 training.

| Parameter | Value |
|---|---|
| Dropout | 0.3 |
| Embedding dim | 512 |
| FFN dim | 2048 |
| Attention heads | 8 |
| Encoder layers | 6/18 |
| Decoder layers | 6/18 |
| Learning rate | $1 * 10^{-3}$ |
| Learning rate scheduler | inverse sqrt |
| Warm-up steps | 4000 |
| Label smoothing | 0.1 |
| Weight decay | 0.0001 |
| Gradient clipping | 0 |
| Adam $\beta$ | 0.9, 0.98 |
| Max update steps | 500k |

Table 10: Hyper-parameters of WMT training

| Parameter | Value |
|---|---|
| Dropout | 0.1 |
| Embedding dim | 512 |
| FFN dim | 2048 |
| Attention heads | 8 |
| Encoder layers | 18 |
| Decoder layers | 18 |
| Learning rate | $1 * 10^{-3}$ |
| Learning rate scheduler | inverse sqrt |
| Warm-up steps | 4000 |
| Label smoothing | 0.1 |
| Weight decay | 0.0001 |
| Gradient clipping | 0 |
| Adam $\beta$ | 0.9, 0.98 |
| Max update steps | 100k |

Table 11: Hyper-parameters of OPUS-100 training

