# OpenReview forum: "ResiDual: Transformer with Dual Residual Connections"
_ICLR.cc/2024/Conference — Submitted to ICLR 2024_

### Official Review · Reviewer_e3RM · 2023-10-29

**Soundness:** 3 good
**Presentation:** 3 good
**Contribution:** 3 good
**Rating:** 6
**Confidence:** 3

**Summary:**

The paper proposes a novel Transformer architecture named ResiDual to explore the optimal way implementing residual connections in Transformer. There are two widely used methods in the literature: Post-Layer-Normalization (Post-LN) and Pre-Layer-Normalization (Pre-LN). However, they suffer from either the gradient vanishing issue or the collapse issue. The paper combines the two LN ideas and designs an architecture with Pre-Post-Layer-Normalization. Basically, the new proposed architecture has two parallel lines, one for Post-LN and the other for Pre-LN. The two lines share the same self-attention block. The authors claim that this architecture can avoid the aforementioned limitations. They first give a theoretical analysis, and then conduct experiments on several data sets. They show that the new architecture can outperform baseline methods on benchmarks, like IWSLT-14, WMT, and OPUS-100.

**Strengths:**

- How to implement residual connections in Transformer optimally is a very important topic in deep learning. The paper makes contributions in this direction, which, in my opinion, should be of interest to the community.

- The paper is overall well-written. The basic idea is clean and easy to understand.

**Weaknesses:**

- The experimental results are a little bit unconvincing. The experiments section did not state the number of algorithm runs used to compute each reported result. Additionally, it is not clear that the results present in the tables are the best result among several runs or the average.

- Only two settings (E6D6 and E12D12) are considered in the experiments. It will be nice to have a figure about how the algorithms' performances vary as the number of layers increases.

**Questions:**

- (1)  Are the experimental results the best among several runs or the average?  If the average, what's the standard deviation?

- (2) Have you tried the structure that simply lets all odd blocks be Post-LN and all even blocks be Pre-LN (or reversed)?

---

### Official Review · Reviewer_ygjB · 2023-10-30

**Soundness:** 3 good
**Presentation:** 3 good
**Contribution:** 3 good
**Rating:** 5
**Confidence:** 3

**Summary:**

This paper points out the shortcomings of the Post-LN and Pre-LN Transformer architectures. Post-LN suffers from the gradient vanishing problem, while Pre-LN has the representation collapse issue. To address the problems of the aforementioned architectures, the paper introduces a new architecture called ResiDual. This architecture combines the advantages of Post-LN and Pre-LN and attempts to avoid their drawbacks. Subsequently, a series of experiments are conducted based on this model.

**Strengths:**

(1) The research focus of this paper is very clear, and the approach is well-organized.
(2) The ResiDual model proposed in the paper indeed improves upon the combination of Post-LN and Pre-LN and achieves good results on some datasets.

**Weaknesses:**

(1) Based on the empirically observed results, the computational cost increased by 3%. Is there sufficient experimental data to support this? I didn't see relevant information in the text or the appendix.
(2) Your model seems to be prone to overfitting. The results on the IWSLT and WMT datasets show that the depth of your model is always limited, which to some extent reduces the potential for performance improvement of the model itself.
(3) When studying learning rate warm-up, you shouldn't use small datasets. Switch to a larger dataset and a deeper network, and your performance improvement will be more convincing.

**Questions:**

Please refer to the weaknesses.

---

### Official Review · Reviewer_EUoZ · 2023-10-30

**Soundness:** 1 poor
**Presentation:** 2 fair
**Contribution:** 1 poor
**Rating:** 1
**Confidence:** 5

**Summary:**

The authors propose a method "Residual", where they accumulate the output of all the layers of a Post-LN transformer model, which is then normalized and added to the output of the model. The author's method behaves similarly to a Post-LN model, while avoid the gradient vanishing problem, as the gradient can flow back freely along the accumulating output. The authors show that vanishing gradient problem of Post-LNs may not be solveable by using Adaptive Optimizers such as Adam, as their update rule becomes ill-conditioned for very very low values of gradient. The authors show their method outperforms other prior method across multiple datasets.

**Strengths:**

1. The authors method avoid the "representation collapse" issue, where the representation changes less for each new layer in Pre-LN.
2. The authors compare their method on multiple datasets

**Weaknesses:**

1. The treatment of grad norm of Pre-LN seems to be sometimes empirically incorrect (see Questions for authors), with wildly different observations in reality compared to the author's theoretical treatment
1. Their derivations of theorem 3.1 ignore the impact of non-linearity on the gradient of the transformer MLP layer, instead assuming it to be a single FC.
1. The derivation also ignores back-propagated gradient through the Query - while Keys will not back-propagate any gradient as Query is zero-initialized, Query will have non-zero gradient back-propagating.
1. As the improvements can sometimes be small (For eg. 27.65 vs 27.3 in Table 3), some measure of statistical significance of the improvements is required.
1. Discussion of prior work is somewhat lacking - while the authors reference prior works throughout the text, a dedicated section for more detailed discussion is missing.
1. The discussion of Adam condition number assumes very very small values of gradient, which is not observed realistically. This discussion should be contextualized given realistic values of the gradient.

**Questions:**

1. Could the authors provide code/provide exact steps to reproduce figure 2(a), specifically the gradient-norm of Pre-LN? I tried to reproduce this figure, and I am failing. See details below on the exact code I used. The plot I observed does not look anything like what Equation(2) would suggest - it seems perhaps exponential for shallow layers, but definitely not logarithmic.
2. For experiments in Table 2, 3, and 4, did the authors only run a single run, or were the experiments repeated multiple times with varying hyper-parameters? How was this hyper-parameter search performed? The dropout and LR (from Table 9,10, 11) are different for these tables.
    1. So if a hyper-parameter search was performed, was the same search performed for all the baselines?
3. Figure 3 in the appendix shows the condition number upto $\sigma_g$ upto $10^{-7}$ - what is the realistic range of $\sigma_g$? On experimenting with standard BERT-large model, I found $\sigma_g$ at initialization was < $10^{-4}$ for 0.3% of params, and even < $10^{-3}$ for 4% of params. How is the Adam condition number at these (somewhat more-realistic) lower values of  $\sigma_g$, and how does Figure 3(left) change?



Code for gradient norm of Pre-LN -
Using a sample LM code from huggingface from [here](https://github.com/huggingface/transformers/blob/v4.31.0/examples/pytorch/language-modeling/run_clm_no_trainer.py), I added ` config.n_layer = 48` at line 390 to increase the number of layers. And then I ran the command below -
```
python run_clm_no_trainer.py  --model_type gpt2 --tokenizer_name gpt2 --dataset_name wikitext  --dataset_config_name wikitext-2-raw-v1 --per_device_train_batch_size 2 --output_dir temp --seed 1234
```
After 1 backward iteration, I plot the grad norm of parameters.


Minor typos (the authors are not expected to respond to these) -
1. Cite published version of Liu et al 2020
1. "So does the gradients of $w_k$ in section 2.1 -> so do
1. "Gradient vanish issue" -> vanishing
1. "the both disadvantages"
1. "non of these models"

---

> ### Author Response · Authors · 2023-11-23
> **response**
>
> Dear EUoZ,
>
> Thank you for your insightful review and the time you have invested in providing your valuable feedback on our paper. We appreciate your careful reading and have addressed each of your comments as follows:
>
> **Q1: The grad norm of Pre-LN and how to reproduce Fig 2(a).**
>
> Our theoretical study is consistent with the findings of previous works, wherein the main conclusion is that Pre-LN Transformer networks do not suffer from the gradient vanishing issue. Regarding the implementation, it is entirely based on FairSeq. To facilitate reproduction, we have updated a notebook in our anonymous repository to generate the grad norm for each layer, which can be found at https://anonymous.4open.science/r/residual_review-6F08/notebooks/vis_grad.ipynb. We have also executed the command you provided, which relies on the huggingface transformers library, and observed no gradient vanishing issue. The variations in the curve as compared to the FairSeq curve may be attributed to implementation details, which we are eager to investigate further.
>
> **Q2: The derivations of theorem 3.1 ignore the impact of non-linearity on the gradient of the transformer MLP layer, instead assuming it to be a single FC.**
>
> First, Theorem 3.1 is to illustrate the gradient norm scale across layers in a pre/post-norm transformer, and the MLP with or without ReLU activation in between does not change conclusion at all because of the piece-wise linear property of ReLU and statistical concentration property of randomly initialized weights. Second, the analysis of the forward and backward processes through ReLU activation is straightforward and readily available in many previous literature (Allen-Zhu & Li 2019 “A convergence theory for deep learning via over-parameterization”, Xiong et al.2020 “On layer normalization in the transformer architecture” and many others). Given such circumstances, we choose to present the result in a concise way without too many sub/super scripts, where we use a simple MLP layer. We have revised the paper with these points clearly stated.
>
> **Q3: The derivation also ignores back-propagated gradient through the Query.**
>
> In our setup, we initialize $W_Q$ to be zero matrices. When doing back-propagation, there is no backward signal on the $xW_K$ branch because of $W_Q=0$. We can compute the gradient on $W_Q$ matrix but there is no backpropagated signal through the $W_Q$ matrix. Hence, our proof and the claim in Theorem 3.1 are flawless as we denote the block k’s parameters to be $w_k$ which includes the MLP layer weights and query matrix. Nonetheless, we will add a note to avoid future misunderstanding.
>
> **Q4: Statistical significance test.**
>
> For baseline model (i.e., B2T), we use the checkpoint they provided on the original authors’ GitHub and generated the output for WMT2010 to WMT2016. We then used Moses tool (https://github.com/moses-smt/mosesdecoder/blob/master/scripts/analysis/bootstrap-hypothesis-difference-significance.pl) to compute the significance value (P-value). The output is:
> Assuming that essentially the same system generated the two hypothesis translations (null-hypothesis), the probability of actually getting them (p-value) is: 0.001.
> Therefore, the null hypothesis, i.e., our system is the same as baseline, should be rejected (p<0.001). In another word, our improvement is statistically significant.
>
> **Q5: More discussion about previous work is missing.**
>
> We appreciate your feedback on this point. In the first section, we have categorized previous works into three groups: modifying architecture, weights, and initialization. However, if there are additional aspects of previous work that need further discussion, we would be grateful if you could provide more details.
>
> **Q6: Adam condition number and values of gradient.**
>
> Regarding Adam, we focused on the vanished gradings in early layers. We have detailed the changes in the $\kappa$ value in the notebook. If a model wider than 1024 (e.g. 4096) or a learning rate larger than 1e-4 (e.g., 5e-4) is used, the $\kappa$ value will be even larger.
>
> **Q7: How hyper parameters are selected.**
>
> One of the advantages of our proposed method is that it does not introduce any new hyper-parameters. The learning rate and dropout are standard settings on these tasks, which can be found in the library tutorial or baseline implementation (e.g., https://github.com/facebookresearch/fairseq/tree/main/examples/translation#iwslt14-german-to-english-transformer or https://github.com/takase/b2t_connection#training). We will provide further clarification on our choice of these specific hyperparameters in the revised version.
>
> **Q8: Typos**
>
> We appreciate you bringing the typographical errors to our attention. We will revise the draft to correct them.
>
>
> We are grateful for your insightful comments and look forward to revising our paper according to your suggestions. We believe that your feedback will significantly improve the quality of our work.
>
> Best regards,

---

> > ### Comment · Reviewer_EUoZ · 2023-11-23
> > **Rebuttal Acknowledgement**
> >
> > Thanks for the response. From a quick skim, the notebook is helpful.
> > If the differences are indeed down to implementation details, it will be interesting to see exactly what causes such large differences. Alternatively, perhaps the differences are due to an encoder-decoder model, vs a decoder only model.
> >
> > I will review the notebook and update my scores accordingly.

---

### Official Review · Reviewer_9QNj · 2023-11-02

**Soundness:** 2 fair
**Presentation:** 3 good
**Contribution:** 2 fair
**Rating:** 6
**Confidence:** 2

**Summary:**

The paper focuses on advancing the Transformer architecture by introducing a novel approach called "ResiDual." This approach aims to address the limitations of two widely used variants by incorporating two residual connections to mitigate gradient vanishing and the representation collapse problem. The paper provides both theoretical analysis and empirical results to validate the effectiveness of the proposed model. The study also delves into the performance of the ResiDual model in various settings, including different datasets like WMT and OPUS-100, and its behavior with different learning rate schedules.

**Strengths:**

About Approach: The introduction of the ResiDual model offers a fresh perspective on addressing the challenges faced by the Transformer architecture.

Comprehensive Experiments: The paper provides extensive experimental results on multiple datasets, showcasing the model's robustness and versatility.

Performance: The ResiDual model show some improvements over other methods.

Stability in Training: The research highlights that the ResiDual model does not require learning-rate warm-up for convergence, unlike some other methods.

**Weaknesses:**

I'm not an expert in NLP and I have limited knowledge on this.

However, one of my concern is about the incrementally performance.

Also, the authors claimed "Post-LN causes gradient vanishing issue that hinders training deep Transformers, and Pre-LN causes representation collapse issue that limits model capacity". Thanks for the theoretical analysis in Sec. 2. However, if the authors could provide more empirical evidence to support that?

**Questions:**

Please see above.

---

### Official Review · Reviewer_q8he · 2023-11-07

**Soundness:** 2 fair
**Presentation:** 3 good
**Contribution:** 3 good
**Rating:** 3
**Confidence:** 3

**Summary:**

The submission introduces a novel Transformer architecture aimed at overcoming the gradient vanishing and representation collapse issues found in Post-LN and Pre-LN models, respectively. The paper provides a theoretical analysis to support the architecture's design and presents empirical results from machine translation tasks across several datasets, demonstrating improvements over existing Transformer variants. The approach combines dual residual connections to retain the benefits of both Post-LN and Pre-LN models.

**Strengths:**

+ The paper is well-organized, with a clear introduction to the problem, a detailed methodology, and a presentation of results that make it accessible to readers.
+ The paper introduces a novel architecture, ResiDual, which creatively combines the benefits of Post-LN and Pre-LN Transformers to address their respective limitations.
+ The submission includes a thorough theoretical examination of the gradient vanishing and representation collapse problems, providing a solid foundation for the proposed solution.

**Weaknesses:**

- Experiments might be restricted to machine translation tasks, which does not demonstrate the model's generalizability across different domains or tasks in NLP or other areas where Transformers are applicable.
- Comparisons with models that have similar enhancements (e.g. RealFormer) to the Transformer architecture, leaving the evaluation incomplete.
- Missing analysis towards computational costs or training efficiency of the ResiDual model.

**Questions:**

1. How does the ResiDual model's computational complexity compare to standard Transformer models, particularly in terms of training time and memory requirements?
2. Can the authors provide additional insights into how the ResiDual model performs on tasks other than machine translation, such as language understanding or speech recognition?
3. Compare the ResiDual model not only with standard Transformer variants but also with recent SOTA models that address similar issues.

---

### Meta-Review · Area_Chair_BfKJ · 2023-12-15

**Metareview:**

The paper attempts to improve transformer architecture. In this regards, the authors propose ResiDual combining the benefits of Post-Layer-Normalization (Post-LN) and Pre-Layer-Normalization (Pre-LN) architectures while avoiding their limitations. The authors provide both theoretical claims and empirical studies on various machine translation benchmarks. While the paper is well-written and the proposal of ResiDual is novel, the reviews highlight several concerns. In particular, unconvincing experimental results, limited settings (experimental and theoretical), missing discussion of prior work, empirical evidence supporting certain theoretical claims. Also the authors did not engage much with reviewers during discussion period to resolve various reviewers concerns, only one response was received. Thus unfortunately, the paper can't be accepted to ICLR in its current form.

**Justification For Why Not Higher Score:**

- Generalizability: Experimental: The paper's focus on machine translation tasks raises questions about its applicability across different domains. Theoretical: Non-linearity effects, range of condition number, convex vs concave etc.

- Comparative Analysis: There is a need for more comprehensive comparisons with models having similar enhancements. like just alternating between the two layer-norms.

- Computational Efficiency: The paper lacks a detailed analysis of the computational costs and training efficiency.

- Empirical Evidence: Some reviewers raised concerns about the empirical evidence supporting certain theoretical claims. In particular reviewer EUoZ ran authors provided notebooks to find some discrepancies.

- Minor Issues: Concerns about overfitting and incremental improvements.

**Justification For Why Not Lower Score:**

N/A

---

### Decision · Program_Chairs · 2024-01-16

Reject